# SPACoder: Alignment-Enhancing Parallel Code Generation for Semi-Supervised Code Translation

## Abstract

Code translation is the task of converting source code from one programming language to another. Sufficient parallel code data is essential for neural code translation models to learn the correct alignment across different languages. However, existing parallel code data is limited in quantity and supported languages. In this paper, we propose a semi-supervised code translation method, **SPACoder**, that leverages snippet training, static analysis, and compilation to generate synthetic parallel code with enhanced alignment in a scalable way, and improves code translation by curriculum learning based on the alignment level of training instances. SPACoder can be generalized to multiple languages and various models with little overhead. Extensive experiments show that SPACoder significantly improves code translation performance on C++, Java, Python, and C, outperforming state-of-the-art baselines by wide margins in execution-based evaluation (CA@1). Notably, we improve C translation by up to 43% with less than 150 annotated training instances.

## 1 Introduction

Code translation is the task of converting source code written in one programming language (PL) to another. This process is valuable for migrating existing code to other languages, and can significantly reduce legacy code maintenance costs and new platform development. Traditional methods for code translation are usually rule-based, which require high expertise in multiple programming languages and considerable manual effort. Benefiting from recent machine learning advances, data-driven methods have shown promising results in automated code translation (Chen et al., 2018; Roziere et al., 2020; 2021b; Zhu et al., 2022). Similar to neural machine translation, neural code translation usually relies on parallel code data for sequence-to-sequence (seq2seq) training. Parallel code data refers to pairs of code snippets from different programming languages that are functionally equivalent and bug-free. Sufficient parallel code data is essential for training models to learn the correct alignment of data structures, APIs, and grammatical rules across different languages. However, existing parallel code data is limited in quantity and supported languages (Nguyen et al., 2013; 2015; Karaivanov et al., 2014; Lu et al., 2021; Chen et al., 2018; Ahmad et al., 2021b; Zhu et al., 2022).

To reduce the dependence on parallel code data, one line of work follows the "pre-training - fine-tuning" approach (Ahmad et al., 2021a; Wang et al., 2021; Roziere et al., 2021a; Fried et al., 2022). Large-scale online code repositories such as GitHub, introduce a vast amount of open source code data. These methods pre-train large language models (LLMs) on open source code with self-supervised learning techniques to gain general knowledge about programming languages, and then fine-tune them on small specialized datasets to perform downstream tasks. Nevertheless, pre-training tasks such as masked language modeling (MLM) (Devlin et al., 2019; Feng et al., 2020; Guo et al., 2020) are usually quite different from the downstream tasks such as code translation, and the performance on the latter is limited by the discrepancy. Another line of work takes an unsupervised learning approach for code translation. Established techniques from unsupervised neural machine translation (NMT) (Lample et al., 2017; Artetxe et al., 2017; Lample et al., 2018; Artetxe et al., 2019), such as back-translation and denoising auto-encoding, can be applied to code data effectively, achieving good performance on code translation without relying on any parallel data (Edunov et al., 2018; Roziere et al., 2020; Agarwal et al., 2021; Ahmad et al., 2022; Szafraniec et al., 2022; Kusum et al., 2022). However, unsupervised learning introduces significant noise in the training process, which is

| Input (**Java**) | Transcoder-ST (**C++**) | SPACoder (**C++**) |
|---|---|---|
| Boolean areElementsContiguous(int arr[], int n) { | Boolean areElementsContiguous (int arr [ ] , int n ) { | bool areElementsContiguous ( int arr [ ] , int n ) { |
|    HashSet<Integer> us = new HashSet<Integer>(); |   HashSet < int > us ; |   unordered_set < int > us ; |
|    for (int i = 0; i < n; i++) |   for ( int i = 0 ; i < n ; i ++ ) { |   for ( int i = 0 ; i < n ; i ++ ) |
|       us.add(arr[i]); |    us . add ( arr [ i ] ) ; |    us . insert ( arr [ i ] ) ; |
|    int count = 1; |   } |   int count = 1 ; |
|    int curr_ele = arr[0] - 1; |   int count = 1 ; |   int curr_ele = arr [ 0 ] - 1 ; |
|    while (us.contains(curr_ele) == true) { |   int curr_ele = arr [ 0 ] - 1 ; |   while (us . find ( curr_ele ) != us . end ( ) ) { |
|       count++; |   while ( us . contains ( curr_ele ) == true ) { |    count ++ ; |
|       curr_ele--; |    count ++ ; |    curr_ele -- ; |
|    } |    curr_ele -- ; |   } |
|    curr_ele = arr[0] + 1; |   } |   curr_ele = arr [ 0 ] + 1 ; |
|    while (us.contains(curr_ele) == true) { |   curr_ele = arr [ 0 ] + 1 ; |   while (us . find ( curr_ele ) != us . end ( ) ) { |
|       count++; |   while ( us . contains ( curr_ele ) == true ) { |    count ++ ; |
|       curr_ele++; |    count ++ ; |    curr_ele ++ ; |
|    } |    curr_ele ++ ; |   } |
|    return (count == (us.size())); |   } |   return ( count == ( us . size ( ) ) ) ; |
| } |   return ( count == ( us . size ( ) ) ) ; | } |
|  | } |  |

Figure 1: An example of the "Shallow Translation" problem, with the Java function shown in the first column as input, the C++ translations from baseline method TransCoder-ST and our proposed method SPACoder (with CodeT5 as generator). The highlighted parts show that TransCoder-ST's translation directly copied types, data structures and statements from the input Java code, which are non-existent or grammatically incorrect in the target language C++, while SPACoder was able to correctly convert them in the corresponding C++ grammar.

particularly harmful to code generation tasks that require precision. Moreover, without training on sufficient parallel code, the self-supervised and unsupervised models can potentially learn incorrect mappings of syntax and data structures from one language to another. For example, they might directly copy tokens and statements from the source language when generating in the target language, or translate the input code token by token and ignore the grammatical rules of the target language. We refer to this issue as "shallow translation". Figure 1 illustrates an example of shallow translation.

Considering the limitations of existing methods, we argue that it is crucial to efficiently generate high-quality and well-aligned parallel code data and effectively learn the cross-lingual alignment. However, programming languages follow rigorous grammatical rules, while neural code generation relies on probabilistic decoding, which is prone to variance and noise. Moreover, it is challenging to generate parallel code for different languages and types of programs in a cost-efficient way, which hurdles parallel code generation at scale. In this paper, we propose a novel **S**emi-supervised code translation method leveraging **P**arallel code generation with enhanced cross-lingual **A**lignment (**SPACoder**). We first generate synthetic parallel code with varying levels of alignment and quality by leveraging snippet training, static code analysis, and compilation. We then train the code translation model on the synthetic and annotated parallel code through alignment-ascending curriculum learning. Compared to existing methods, SPACoder has several advantages. First, the compilation and static analysis secure the syntactical correctness and alignment of the synthetic parallel code in a cost-efficient way; Second, the alignment-ascending curriculum learning is robust to noise in the data, which effectively reduces the shallow translation problem. Moreover, our method can be applied to different languages and types of programs with little overhead, which enables parallel code synthesis at scale. Extensive experiments show that the synthetic parallel code significantly improves the performance of code translation for multiple languages, and is particularly effective for low-resource language.

Our contributions can be summarized as follows: **(1)** We propose a novel semi-supervised code translation method, SPACoder, that leverages snippet training, static analysis, and compilation to generate synthetic parallel code with enhanced alignment in a scalable way. SPACoder can be generalized to multiple languages and various models with little overhead. **(2)** We introduce alignment-ascending curriculum learning, where the code translation model is trained on both synthetic parallel code and annotated parallel code, considering the alignment level, noise level, and quantity of each type of data. We demonstrate that curriculum learning improves the code translation model's performance and enhances alignment across different languages, resulting in more precise translations. **(3)** We evaluate SPACoder with two different underlying models and over 1 million synthetic parallel code pairs across 4 languages, split into different datasets by the level of quality and alignment. Extensive experiments show that SPACoder successfully improves code translation performance by up to 30% on C++, Java, and Python, outperforming state-of-the-art baselines on translation between Python and C++ by 5.7%, C++ and Python by 6%, and Python and Java by 8%

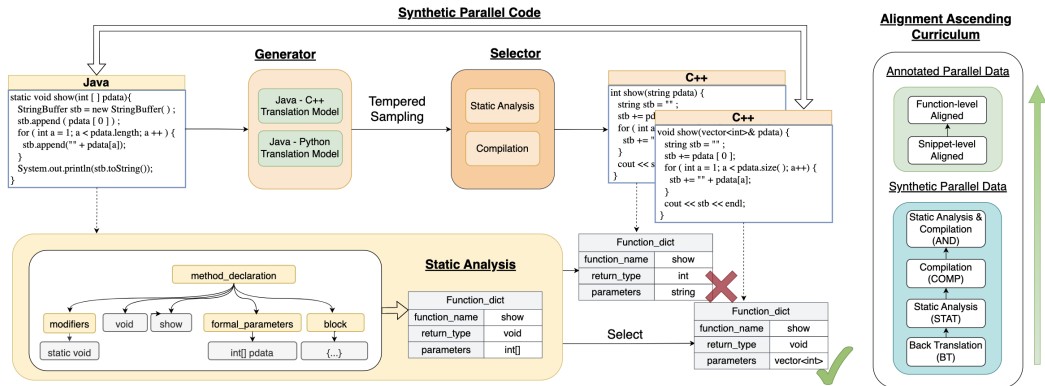

Figure 2: Overview of SPACoder for Code Translation. SPACoder utilizes a two-step process to generate high-quality translation hypotheses from monolingual code inputs. First, the generator produces multiple translation hypotheses using tempered sampling. Next, the selector applies static analysis and compilation techniques to select the most promising hypotheses. By employing various selection criteria, SPACoder generates synthetic parallel code datasets with varying alignment levels and quality. These synthetic datasets, along with annotated parallel code datasets, are organized into a curriculum, where the alignment and quality gradually improve. The proposed curriculum-based approach enhances code translation performance.

in execution-based evaluation (CA@1). Notably, our method improves C translations by up to 43% with less than 150 annotated training instances.

## 2 METHOD

The lack of parallel code data poses a challenge for training code translation models, which rely on large amounts of parallel data to achieve good performance. Semi-supervised methods can leverage monolingual data to generate synthetic parallel data but often struggle to maintain alignment quality between the source and target languages. In this paper, we focus on function-level code translation, as functions are the building blocks of programs. Figure 2 shows an overview of the proposed method.

### 2.1 PARALLEL CODE DATA GENERATION

To address the data scarcity challenge, we propose a parallel code generation method using semi-supervised learning. The method consists of two modules, a hypotheses generator $f_G$, and a selector $f_D$. The hypotheses generator $f_G$ is sequence-to-sequence model that takes as input a code snippet $x$ from the source language $s$ and generates a set of hypothetical translations $\mathcal{Y}_h = \{y_h^{(1)}, y_h^{(2)}, .., y_h^{(M)}\}$ in the target language $t$. Here, $\mathcal{Y}_h$ consists of $M$ translations (hypotheses) for the same input code snippet $x$. The generator $f_G$ is trained on a limited amount of parallel code data ($D_L$, $L$ is for labeled), and will be used to generated a large amount of hypotheses for monolingual code data ($D_U$, $U$ is for unlabeled). The selector $f_D$ comprises a set of $K$ filtering criteria $\mathcal{F} = \{F_k\}_{k=1}^{K}$ where $\widetilde{\mathcal{Y}}_{h,k} = F_k(\mathcal{Y}_h)$ takes $\mathcal{Y}_h$ as input and outputs the subset of hypotheses $\widetilde{\mathcal{Y}}_{h,k} \subset \mathcal{Y}_h$ that passes the criterion $F_k$.

#### 2.1.1 HYPOTHESES GENERATION

The hypotheses generator $f_G$ is initialized by training on a limited amount of parallel code data. This is to enable $f_G$ with the ability of translating code from the source language $s$ to the target language $t$. To further improve $f_G$'s translation capability, we leverage the snippet training method from Zhu et al. (2022), which matches code comments in parallel programs to get snippet-level parallel training data. A snippet usually consists of several lines of code and is not necessarily a complete function. We then use the trained $f_G$ to generate hypotheses for a large amount of monolingual code.

**Snippet Training.** We use two small annotated parallel code datasets, $\mathcal{D}_{L_s}$ and $\mathcal{D}_L$, with different levels of alignment to train $f_G$. The parallel code data aligned at snippet-level is denoted as $\mathcal{D}_{L_s} = \{(x, y)^{(ls)}\}_{ls=1}^{|\mathcal{D}_{L_s}|}$, and the function-level parallel data is denoted as $\mathcal{D}_L = \{(x, y)^{(l)}\}_{l=1}^{|\mathcal{D}_L|}$. $\mathcal{D}_{L_s}$ can be constructed from $\mathcal{D}_L$ by matching code comments from the parallel programs (Zhu et al., 2022). We first train $f_G$ on $\mathcal{D}_{L_s}$, and then continue the training on $\mathcal{D}_L$. We refer to this step as snippet training, which helps the generator to learn fine-grained alignment between different languages and substantially improves $f_G$'s ability to generate hypotheses with better alignment to the input code. This step enables $f_G$ to generate valid hypotheses with sufficient initial quality.

**Tempered Sampling.** Let $\mathcal{D}_U = \{x^{(i)}\}_{i=1}^{|\mathcal{D}_U|}$ be a monolingual dataset in the source language $s$, where each $x^{(i)}$ is a function-level code block. With $\mathcal{D}_U$ as input, we can generate a set of translation hypotheses in the target language $t$ with the trained $f_G$. To increase the diversity of the hypotheses and improve coverage for different possible translations, we use tempered sampling to acquire $M$ ($M$ is the sample size) different hypotheses for each input code. Tempered sampling makes use of a tuned scaled softmax to control the degree of randomness (temperature) in the sampling process (Ackley et al., 1985; Hinton et al., 2015). We denote the hypotheses set as $\mathcal{H} = \{\mathcal{Y}_h^{(1)}, \mathcal{Y}_h^{(2)}, \ldots, \mathcal{Y}_h^{(i)}, \ldots, \mathcal{Y}_h^{|\mathcal{D}_U|}\}$, where $\mathcal{Y}_h^{(i)} = \{y_h^{(1)}, y_h^{(2)}, .., y_h^{(M)}\}$ is a set of different translations for $x_i$ in the target language $t$.

### 2.1.2 Hypotheses Selection

The selector $f_D$ takes $\mathcal{H}$ as input and outputs $\widetilde{\mathcal{H}} = \{\widetilde{\mathcal{Y}}_h^{(i)}\}_{i=1}^{|\mathcal{D}_U|}$, in which $\widetilde{\mathcal{Y}}_h^{(i)}$ is the subset of $\mathcal{Y}_h^{(i)}$ that passes the selection criteria $\mathcal{F}$, i.e., $\widetilde{\mathcal{Y}}_h^{(i)} = \mathcal{F}(\mathcal{Y}_h^{(i)})$. If $\widetilde{\mathcal{Y}}_h^{(i)}$ contains more than one hypothesis, only one is kept, as our preliminary experiments confirm that keeping more than one hypothesis for each input does not yield improved performance [1]. We pair all the $y_h^{(i)}$ with the input corresponding input code $x^{(i)}$ to acquire pseudo parallel dataset $\mathcal{D}_S = \{(x, y_h)^{(l)}\}_{l=1}^{|\mathcal{D}_S|}$. In practice, we rely on cross-lingual static code analysis and compilation as selection criteria $\mathcal{F}$ for the hypotheses.

**Cross-Lingual Static Analysis.** To ensure that the selected hypotheses have high alignment quality with the input code, we use cross-lingual static analysis to compare the key information of both the input code and all the hypotheses. Static code analysis is a technique used to analyze source code without executing the program. One way to perform static code analysis is through the use of an abstract syntax tree (AST). An AST is a tree-like data structure that represents the structure of a program's source code. It captures the high-level structure of the code and the relationships between its elements, enabling a deeper understanding of the code beyond the sequence-level. Figure 2 shows an example AST generated from a Java function.

Specifically, we compare the number of functions, and after matching each pair of functions from output with the input, we check whether the return types are equivalent, and if the parameter lists match in terms of the number of parameters and the type of each parameter. For non-typed languages such as Python, we skip the type part and only compare the number of functions and the parameter list of each function. Passing the cross-lingual static analysis is a strong indicator of the alignment quality of the hypotheses to the input, which helps in selecting the best hypotheses.

**Compilation Filtering.** We also leverage compilation to filter out hypotheses that contain errors. Specifically, we compile the generated code using the target compiler and check for any compilation errors. If a hypothesis fails to compile, it is discarded. This step further improves the quality of the selected hypotheses by ensuring that they are syntactically correct and can be compiled successfully.

### 2.2 Alignment-Ascending Curriculum Learning

By pairing the hypotheses with their corresponding inputs, we obtain multiple synthetic parallel code datasets at different stages of the generation process. Without the selector, the generation is reduced to plain back-translation. We denote the unfiltered synthetic parallel data from the unfiltered hypotheses, as BT data. Similarly, we denote the synthetic parallel data from cross-lingual static analysis and compilation filtering as STAT and COMP, respectively. In addition, we denote the subset of hypotheses that pass both criteria, static analysis and compilation, as AND data. We adopt a

---

[1]If $\widetilde{\mathcal{Y}}_h^{(i)}$ is empty, it will be discarded.

| Input (Java) | Synthetic Output (C++) | Input (Python) | Synthetic Output (C++) |
|---|---|---|---|
| ```static void show(int [ ] pdata){   StringBuffer stb = new StringBuffer( ) ;   stb.append ( pdata [ 0 ] ) ;   for ( int a = 1; a < pdata.length; a ++ ) {     stb.append("" + pdata[a]);   }   System.out.println(stb.toString()); }``` | ```void show(vector<int>& pdata) {   string stb = "" ;   stb += pdata [ 0 ];   for ( int a = 1; a < pdata.size( ); a++) {     stb += "" + pdata[a];   }   cout << stb << endl; }``` | ```def f ( a , b ) :   p , q = a , b   while q != 0 :     p , q = q , p % q   return a / p * b``` | ```float f ( float a , float b ) {   int p = a , q = b ;   while ( q != 0 ) {     p = q ;     q = p % q ;   }   return a / p * b ; }``` |
| ```public static long gcd ( long a , long b ){   long c = a % b ;   if ( c == 0 ) {     return b ;   }   else {     return gcd ( b , c ) ;   } }``` | ```long gcd ( long a , long b ) {   long c = a % b ;   if ( c == 07 )     return b ;   else     return gcd ( b , c ) ; }``` | ```def check ( n , array ) :   if 1.1 <= n :     array [ 0 ] += 1   elif 0.6 <= n < 1.1 :     array [ 1 ] += 1   elif 0.2 <= n < 0.6 :     array [ 2 ] += 1   else :     array [ 3 ] += 1``` | ```void check ( int n , int array [ ] ) {   if ( 1.0 <= n )     array [ 0 ] ++ ;   else if ( 1.0 <= n )     array [ 1 ] ++ ;   else if ( 0.2 <= n )     array [ 2 ] ++ ;   else   array [ 3 ] ++ ; }``` |

Figure 3: Synthetic parallel code examples from CODENET-SPACoder, with PLBART (Ahmad et al., 2021a) as generator. The synthetic parallel data demonstrates great alignment quality, with minor noise in some cases.

curriculum learning approach to train our code translation model, strategically leveraging the quality of the data at different stages. Our curriculum consists of multiple training phases, progressively incorporating different types of data. We first train with the unfiltered synthetic parallel data, allowing the model to grasp the basic translation patterns. Next, we introduce the cross-lingual static analysis filtered data, which helps refine the model's understanding of language-specific code idioms and improve translation accuracy. Subsequently, we integrate the compilation filtered data, which further enhances the model's ability to generate syntactically correct translations. The curriculum then advances to utilize the intersection of both filtered datasets, combining the benefits of both data sources. We then introduce snippet-level annotated data to enhance translation performance in specific code segments. Finally, we conclude by training with function-level annotated data, enabling the model to capture higher-level structural patterns and produce more coherent translations. By following this carefully designed curriculum, SPACoder not only benefits from exposure to a diverse range of training data but also progressively refines its translation quality and alignment, leading to improved performance and robustness.

## 3 EXPERIMENTS

**Datasets.** We make use of the annotated CoST dataset from Zhu et al. (2022) to support snippet training and execution-based evaluation. The CoST dataset contains parallel code aligned at both program and snippet levels. To support execution-based evaluation, we execute all programs in CoST and remove the ones that throw run-time errors and the ones with empty execution output. We refer to the resulting dataset as ECoST (Execution-based CoST). ECoST has approximately $1,000$ function-level training instances for C++, Java, and Python, and $150$ for C. We employ a train/validation/test split ratio of approximately 70:5:25. To support snippet and function-level training, we extract the functions from ECoST through AST parsing[2] to get both snippet-level and function-level parallel data ($\mathcal{D}_{L_s}$ and $\mathcal{D}_L$), which we refer to as ECoST-snippet and ECoST-function.

**Synthetic Parallel Code Generation.** We use the CODENET dataset (Puri et al., 2021) as the monolingual code data ($\mathcal{D}_U$) for parallel code generation. CODENET is a large-scale dataset containing 13M programs spanning 55 languages. The programs in CODENET originate from code submissions to online judge of programming problems. We select the "Accepted" submissions (*i.e.*, submissions that pass the online judge) in 4 languages, C++, Java, Python and C, from around $1,600$ problems. After some quality filtering, we get approximately $87,000$ examples. We experiment with two different models as the generator model, PLBART (Ahmad et al., 2021a) and CodeT5 (Wang et al., 2021). The monolingual CODENET data are used as inputs to the generators to obtain the hypotheses through tempered sampling with a temperature of 0.5 and sample size ($M$) of 10. We then get the synthetic parallel code through selection by static analysis and compilation ($\mathcal{F}$). Although we use CODENET as a source for monolingual code, by pairing submissions for the same problem in different languages, one can also get parallel code from CODENET. We use the parallel code from

---

[2]https://tree-sitter.github.io/tree-sitter/

| PLBART | Number of Pairs | | | | | | Selection Rate | | | | | |
|---|---|---|---|---|---|---|---|---|---|---|---|---|
| Selector | C++ – Java | C++ – Py | C++ – C | Java – Py | Java – C | Py – C | C++ – Java | C++ – Py | C++ – C | Java – Py | Java – C | Py – C |
| Back Translation (BT) | 47540 | 63637 | 49550 | 37233 | 22919 | 39231 | 1 | 1 | 1 | 1 | 1 | 1 |
| Static Analysis (STAT) | 25211 | 58157 | 14945 | 31228 | 13059 | 33882 | 0.53 | 0.91 | 0.30 | 0.84 | 0.57 | 0.86 |
| Compilation (COMP) | 15258 | 36224 | 1893 | 13525 | 1562 | 11088 | 0.32 | 0.57 | 0.04 | 0.36 | 0.07 | 0.28 |
| SA & Compilation (AND) | 9278 | 34733 | 1200 | 12104 | 1313 | 10730 | 0.20 | 0.55 | 0.02 | 0.33 | 0.06 | 0.27 |

Table 1: Statistics of SPACoder-function, with PLBART (Ahmad et al., 2021a) as generator. Due to page limit, statistics for CodeT5 (Wang et al., 2021) generated data are included in Appendix. SA & Compilation refers to the intersection of the Static Analysis and Compilation selections.

| PLBART | CodeBLEU | | | | | | Computation Accuracy | | | | | |
|---|---|---|---|---|---|---|---|---|---|---|---|---|
| Dataset | Java – C++ | Py – C++ | C++ – Java | Py – Java | C++ – Py | Java – Py | Java – C++ | Py – C++ | C++ – Java | Py – Java | C++ – Py | Java – Py |
| CODENET-parallel | 21.61 | 21.70 | 22.14 | 19.16 | 20.31 | 18.92 | 0 | 3.13 | 0.64 | 1.57 | 3.37 | 0 |
| ECOST-function | 38.87 | 53.60 | 41.42 | 46.24 | 53.94 | 50.50 | 0.81 | 4.52 | 1.88 | 3.63 | 16.87 | 16.62 |
| ECOST-snippet&function | **71.39** | **66.62** | **71.27** | **64.76** | **62.05** | 60.62 | 25.54 | 24.40 | 27.15 | 23.87 | 32.23 | 32.33 |
| **SPACoder-function** | 69.02 | 65.92 | 70.96 | 63.54 | 61.77 | **61.52** | **38.44** | **27.71** | **28.49** | **24.17** | **35.54** | **37.76** |

Table 2: Performance comparison of the same model trained on existing parallel code data versus on SPACoder-function. The model used here is PLBART. The results from training on SPACoder-function demonstrate superior Computation Accuracy over existing parallel code data, indicating its high quality and effectiveness in improving code translation. Py is short for Python. ECOST-snippet&function means the combination of ECOST-snippet and ECOST-function.

CODENET, denoted as CODENET-parallel, in Table 2 as a dataset baseline to compare the quality of the generated parallel code.

**Baselines and Evaluation Metrics.** We compare against five advanced code translation models. CodeBERT (Feng et al., 2020), PLBART (Ahmad et al., 2021a), and CodeT5 (Wang et al., 2021) are programming language models pre-trained with self-supervised learning techniques on large-scale open-source code datasets. These models can perform code translation as a downstream task after fine-tuning on parallel code data. TransCoder (Roziere et al., 2020) is an unsupervised code translation model that relied on back-translation for data augmentation. TransCoder-ST (Roziere et al., 2021b) improves TransCoder by leveraging unit testing to generate parallel code data. After generating the synthetic parallel code, we train code translation models using the generated data and evaluate their performances. CodeBERT, PLBART and CodeT5 need fine-tuning to perform code translation, therefore they are fine-tuned on ECOST with both snippet-level and function-level data. On the other hand, TransCoder and TransCoder-ST do not need fine-tuning as they are unsupervised methods. All models are evaluated on ECOST test set. CodeBLEU(Ren et al., 2020) is a weighted sum of n-gram matching, AST matching, and data flow matching between source and target programs. Computation Accuracy (CA) (Roziere et al., 2020) is a new metric introduced in TransCoder that measures if the hypothesis has the same execution output as the reference. We use CA@1 for all the evaluations. Model training details are included in the Appendix.

## 4 RESULTS AND ANALYSIS

We evaluate two variations of our method, SPACoder-PLBART and SPACoder-CodeT5, by performing parallel code generation with PLBART and CodeT5 as generators and curriculum learning with their generated data respectively. The generated parallel code data is referred as SPACoder-function. We focus on two aspects, generated data quality and improvements in code translation performance.

### 4.1 QUALITY OF THE SYNTHETIC PARALLEL CODE

**Statistics of SPACoder-function.** With $86,972$ monolingual code as input, we manage to generate $516,142$ and $529,108$ synthetic parallel code pairs in 6 language pairs from PLBART and CodeT5, respectively. Table 1 shows the statistics of the synthetic parallel code data generated by PLBART. Note that the datasets resulting from static analysis and compilation are not subsets of back-translation, because for back-translation we randomly pick a hypothesis from the 10 sampled hypotheses, and for static analysis and compilation we select the hypothesis from the ones that pass the selection criteria. From the selection rate, we can observe that static analysis is the most lenient to Python, as it is a

| Model | CodeBLEU | | | | | | Computation Accuracy | | | | | |
|---|---|---|---|---|---|---|---|---|---|---|---|---|
| | Java – C++ | Py – C++ | C++–Java | Py – Java | C++ – Py | Java – Py | Java – C++ | Py – C++ | C++ – Java | Py – Java | C++ –Py | Java – Py |
| CodeBERT | 61.75 | 50.18 | 29.71 | 42.21 | 46.99 | 46.69 | 13.44 | 4.82 | 10.22 | 3.93 | 6.33 | 5.74 |
| PLBART | 71.39 | 66.62 | 71.27 | 64.76 | 62.05 | 60.62 | 25.54 | 24.40 | 27.15 | 23.87 | 32.23 | 32.33 |
| CodeT5 | 72.76 | 64.99 | 72.13 | 64.26 | 59.16 | 61.25 | 37.63 | 19.28 | 41.13 | 23.87 | 20.78 | 24.77 |
| Trancoder | 72.54 | 66.47 | 70.36 | 63.61 | 56.29 | 55.29 | 49.73 | 25.60 | 40.86 | 22.36 | 41.87 | 46.22 |
| Trancoder-ST | 71.47 | 61.28 | 70.96 | 64.81 | 58.85 | 57.70 | 51.08 | 36.14 | 44.09 | 35.35 | 43.98 | **51.96** |
| **SPACoder-PLBART** | 74.55 | 68.43 | 72.90 | 67.14 | 63.09 | 63.47 | 41.94 | 35.24 | 40.05 | 33.84 | 38.55 | 41.09 |
| **SPACoder-CodeT5** | **74.94** | **69.25** | **74.85** | **69.64** | **65.10** | **65.95** | **51.08** | **41.87** | **49.19** | **43.20** | **50.00** | 49.55 |

Table 3: Performance comparison of two implementations of SPACoder with PLBART and CodeT5 against baseline approaches. The metrics used for comparison are CodeBLEU and Computation Accuracy (CA@1). Across both measures, SPACoder outperforms the baseline approaches, demonstrating its effectiveness in code translation.

| Model | CodeBLEU | | | | | | Computation Accuracy | | | | | |
|---|---|---|---|---|---|---|---|---|---|---|---|---|
| | C++ – C | Java–C | Python – C | C – C++ | C–Java | C – Python | C++ – C | Java – C | Python – C | C – C++ | C – Java | C – Python |
| PLBART | 40.66 | 56.85 | 43.66 | 42.77 | 32.49 | 52.98 | 2.60 | 0 | 1.56 | 5.19 | 0 | 14.06 |
| **SPACoder-PLBART** | **79.08** | **72.37** | **61.73** | **80.34** | **68.79** | **61.92** | **33.77** | **28.77** | **17.19** | **48.05** | **23.29** | **28.12** |
| CodeT5 | 82.06 | 74.16 | 62.25 | 80.04 | 71.25 | 61.06 | 66.23 | 47.95 | 25.00 | 64.94 | 39.73 | 28.12 |
| **SPACoder-CodeT5** | **82.26** | **74.59** | **63.87** | **81.24** | **74.21** | **66.65** | **68.83** | **56.16** | **31.25** | 64.94 | **45.21** | **51.56** |

Table 4: Performance comparison before and after applying SPACoder on low-resource language C. The results show substantial performance improvements across all measures after the application of our method, indicating the effectiveness of SPACoder on low-resource languages.

| Model | CodeBLEU | | | | | | Computation Accuracy | | | | | |
|---|---|---|---|---|---|---|---|---|---|---|---|---|
| | Java – C++ | Py – C++ | C++ – Java | Py – Java | C++ – Py | Java – Py | Java–C++ | Py – C++ | C++ – Java | Py – Java | C++ – Py | Java – Py |
| Base Model | 38.87 | 53.6 | 41.42 | 46.24 | 53.94 | 50.50 | 0.81 | 4.52 | 1.88 | 3.63 | 16.87 | 16.62 |
| BT | 67.91 | 66.32 | 70.54 | 65.50 | 61.78 | 62.40 | 19.62 | 23.8 | 32.26 | 27.19 | 35.24 | 37.76 |
| BT + STAT | 74.35 | 68.23 | 71.56 | 66.19 | 62.45 | 62.83 | 38.71 | 28.92 | 34.95 | 29.31 | 36.45 | 38.97 |
| BT + STAT + COMP | 74.22 | 68.12 | 72.26 | 66.20 | 62.13 | 62.11 | 40.59 | 34.34 | 36.02 | 28.10 | 35.54 | 38.97 |
| **SPACoder** | **74.55** | **68.43** | **72.90** | **67.14** | **63.09** | **63.47** | **41.94** | **35.24** | **40.05** | **33.84** | **38.55** | **41.09** |

Table 5: SPACoder ablation study, showing the curriculum performance improvements in code translation as each synthetic parallel code dataset is added to the alignment-ascending curriculum. The results demonstrate the cumulative contribution of each synthetic dataset to enhancing the effectiveness of the training curriculum. Base model is PLBART trained on ECoST-function.

weakly-typed language. Compilation has the least selection rate on C. This is due to data scarcity as the generator has poor performance on C due to being trained with less than 150 examples.

**Quantitative Analysis.** We quantitatively evaluate the quality of the synthetic parallel code data, SPACoder-function. Specifically, we train PLBART using only the generated data from SPACoder-PLBART and compare it with PLBART trained on other existing parallel data. Table 2 shows the performance from training on each dataset. We observe that on Computation Accuracy, SPACoder-function from PLBART outperforms the annotated datasets on all the language pairs, which is a strong indicator of its high quality and good alignment. CODENET has the worst performance, with close to zero on Computation Accuracy, which is potentially due to its poor alignment quality.

**Qualitative Analysis.** We further perform qualitative analysis and manually inspect samples of the generated data. Table 3 illustrates four examples from the synthetic parallel code, with two in Java – C++, and two in Python – C++. The Java and Python codes are the monolingual input from CODENET, and the C++ codes are the synthetic codes. The generated code snippets are in good alignment with their corresponding inputs, with correct mapping of types, data structures, and syntax. Note that the synthetic codes still contain some noise, for example, there are some mistranslations of the numbers from the input code. However, Table 3 and 4 results indicate that it does not impede the effectiveness of the synthetic code in improving code translation performance.

## 4.2 IMPROVEMENT IN CODE TRANSLATION PERFORMANCE

**Comparison with Baseline Models.** Table 3 shows the CodeBLEU and Computation Accuracy performance on C++, Java, and Python of the baseline models and SPACoder-PLBART and SPACoder-

| Input (C++) | SPACoder (Python) | Transcoder-ST (Python) | CodeT5 (Python) | Groundtruth (Python) |
|---|---|---|---|---|
| int getMin(int arr[], int n) {
    return *min_element(arr, arr + n); }
int getMax(int arr[], int n) {
    return *max_element(arr, arr + n); } | def getMin ( arr , n ) :
  return min ( arr )
def getMax ( arr , n ) :
  return max ( arr ) | def getMin ( arr , n ) :
  return * min_element ( arr , n ) | def getMin ( arr , n ) :
  return min_element (arr, arr + n )
def getMax ( arr , n ) :
  return max_element (arr, arr + n
) | def getMin(arr,n):
  return min(arr)
def getMax(arr,n):
  return max(arr) |

| Test Code (Python) | Execution Output | | | |
|---|---|---|---|---|
| arr = [12,1234,45,67,1]
n = len(arr)
getMin(arr, n); getMax(arr, n) | 1
1234 | SyntaxError: can't use starred expression here | NameError: name 'min_element' is not defined | 1
1234 |

| Input (Python) | SPACoder (Java) | Transcoder-ST (Java) | CodeT5 (Java) | Groundtruth (Java) |
|---|---|---|---|---|
| def nextPowerOf2(n):
  count = 0;
  if (n and not (n & (n - 1))):
    return n
  while( n != 0):
    n >>= 1
    count += 1
  return 1 << count; | static int nextPowerOf2 ( int n ) {
  int count = 0 ;
  if ( n != 0 && ( n & ( n - 1 ) ) == 0 )
    return n ;
  while ( n != 0 ) {
    n >>= 1 ;
    count ++ ;
  }
  return ( 1 << count ) ;
} | public static int nextPowerOf2 ( int n ) {
  int count = 0 ;
  if ( ( n != 0 ) && ( ! ( n & ( n - 1 ) ) ) ) {
    return n ; }
  while ( ( n != 0 ) ) {
    n >>= 1 ;
    count ++ ;
  }
  return 1 << count ;
} | static int nextPowerOf2 ( int n ) {
  int count = 0 ;
  if ( n != 0 && ! ( n & ( n - 1 ) ) )
    return n ;
  while ( n != 0 ) {
    n >>= 1 ;
    count ++ ;
  }
  return ( 1 << count ) ;
} | static int nextPowerOf2(int n) {
  int count = 0;
  if (n > 0 && (n & (n - 1)) == 0)
    return n;
  while(n != 0) {
    n >>= 1;
    count += 1;
  }
  return 1 << count;
} |

| Test Code (Java) | Execution Output | | | |
|---|---|---|---|---|
| public static void main(String args[]) {
  int n = 0;
  System.out.println(nextPowerOf2(n));
} | 1 | error: bad operand type int for unary operator '!' | error: bad operand type int for unary operator '!' | 1 |

| Input (Java) | SPACoder (Python) | Transcoder-ST (Python) | CodeT5 (Python) | Groundtruth (Python) |
|---|---|---|---|---|
| static int findTrailingZeros(int n) {
  int count = 0;
  for (int i = 5; n / i >= 1; i *= 5)
    count += n / i;
  return count;
} | def findTrailingZeros ( n ) :
  count = 0
  i = 5
  while n // i >= 1 :
    count += n // i
    i *= 5
  return count | def findTrailingZeros ( n ) :
  count = 0
  for i in range ( 5 , n // i + 1 , 5 ) :
    count += n // i
  return count | def findTrailingZeros ( n ) :
  count = 0
  for i in range ( 5 ) :
    count += n // i
  return count | def findTrailingZeros(n):
  count = 0
  while(n >= 5):
    n //= 5
    count += n
  return count |

| Test Code (Python) | Execution Output | | | |
|---|---|---|---|---|
| n = 100
print("Count of trailing 0s " +
"in 100! is", findTrailingZeros(n)) | Count of trailing 0s in 100! is 24 | UnboundLocalError: local variable 'i' referenced before assignment | ZeroDivisionError: integer division or modulo by zero | Count of trailing 0s in 100! is 24 |

Figure 4: Qualitative translation results from SPACoder and the baseline methods given the same input. In all three examples, the baselines' results exhibit the "Shallow Translation" problem, where code snippets are directly copied or translated token by token from the source language, causing compilation and run-time errors in the target language. SPACoder's translation shows its strong ability in correctly aligning the syntax and APIs across different languages.

CodeT5. In terms of CodeBLEU, both SPACoder models outperform all baselines, with SPACoder-CodeT5 surpassing the best baseline performance by $8\%$ on Python – C++ and Java – Python translation. In terms of Computation Accuracy, SPACoder-CodeT5 outperforms the best baseline performance by $5\%$ on Python – C++ and C++ – Java, $6\%$ on C++-Python, and $8\%$ on Python-Java. Moreover, both SPACoder models outperform their respective generator models on all the language pairs and both metrics by a wide margin. Compared to CodeT5, SPACoder-CodeT5's Computation Accuracy on Python – C++ and Python – Java improves by $20\%$, and on Java – Python and C++ – Python the improvements are $25\%$ and $30\%$, respectively.

**Performance on Low-resource Languages.** In ECoST, C only has less than $150$ parallel code pairs with each language, making it suitable for evaluating in more challenging low-resource language settings. As shown in Table 1, the compilation rate is the lowest when C is involved, as the generator is not able to generate high-quality data when the training data of C is significantly less. Table 4 shows the performance of the two implementations of SPACoder and their respective generators. For PLBART, SPACoder improves the CodeBLEU by up to $40\%$ and improves the Computation Accuracy by up to $43\%$. This shows that the augmentation of parallel code generation works well in low-resource language settings, where the generator's performance is weak. For CodeT5, the improvement in Computation Accuracy is up to $23\%$.

**Impact of Curriculum on Translation Performance.** Table 5 shows the results of an ablation study designed to incrementally add each synthetic and annotated parallel code dataset to the alignment-ascending curriculum used for training the code translation model. Starting with a base model trained solely on the annotated dataset ECoST (function-level), we progressively add each dataset one by one into the curriculum. The results demonstrate that each added synthetic dataset enhances the model's performance on both metrics. Notably, the best performance is achieved when all synthetic and

annotated datasets are included in the curriculum (SPACoder), underlining the cumulative contribution of each dataset in the curriculum.

**Qualitative Analysis.** Figure 4 shows examples of various model translations and their execution outputs given the same input code. The first column corresponds to the code used as input in the source language, and the last column corresponds to the ground truth translation in the target language. All examples are from the ECoST test set. We compare SPACoder-CodeT5 with two other baselines, TransCoder-ST and CodeT5. In the first two examples, we observe that both baselines demonstrate the "shallow translation" problem. In the C++ – Python example, both TransCoder-ST and CodeT5 directly copy from the input code. While `min_element` is a valid built-in function defined in header `<algorithm>` in C++, it does not exist in Python, resulting in compilation errors for both baselines. TransCoder-ST also exhibits inability in translating multiple functions at once. In the Python – Java example, both TransCoder-ST and CodeT5 translate the keyword `"not"` in Python to `"!"` in Java. However, operator `"!"` cannot be used when the operand is an integer. By translating at token level, these baselines fail in taking the context into consideration, causing run-time errors. In both cases, SPACoder-CodeT5 is able to translate the function calls and statements from the source language to the target language correctly. In the Java – Python example, both baselines fail at translating a complex `for` loop, while SPACoder correctly translates this in a different way from the ground truth, showing a strong capability of understanding the input code and mapping it into a different language.

## 5 RELATED WORK

Recent advances in machine learning, especially in self-supervised learning techniques, have benefited a wide range of tasks (Vaswani et al., 2017; Liu et al., 2019; Lample & Conneau, 2019; Liu et al., 2020; Sehwag et al.). Some techniques from NLP were transferred to programming languages and have achieved great success. Similar to BERT (Devlin et al., 2019), CodeBERT (Feng et al., 2020) is a code language model pre-trained on CodeSearchNet (Husain et al., 2019) with Masked Language Modeling (MLM). PLBART (Ahmad et al., 2021a) is pre-trained the same way as BART (Lewis et al., 2020), with Denoising Auto-Encoding (DAE) (Lample et al., 2018) on GitHub data. Although CodeBERT and PLBART are pre-trained on code, they model code the same way as natural language sequences without taking code-specific features into consideration. Inspired by T5 (Raffel et al., 2020), CodeT5 (Wang et al., 2021) is pre-trained on CodeSearchNet but with an identifier-aware objective to align more with programming language distributions. All three models use general pre-training to gain programming language intelligence, without optimizing for any specific tasks. They require fine-tuning on task-specific data to perform downstream tasks. TransCoder (Roziere et al., 2020) is an unsupervised code translation model that relies on back-translation to generate pseudo-parallel code data during training. However, back-translation introduces noisy code into the training process, compromising the model's ability to generate high-quality translations. TransCoder-ST (Roziere et al., 2021b) improves TransCoder by adding automated unit tests to filter out invalid translations and reduce noise from the back-translation process. However, obtaining unit tests for different languages is expensive, and running unit tests is unscalable for a large amount of code data. MuST-PT (Zhu et al., 2022) leverages snippet-level DAE and translations for pre-training before fine-tuning on program-level data, which improves code translation performance. However, MuST-PT relies solely on limited amount of finely aligned parallel code for training without utilizing widely available non-parallel code, which makes this method less scalable.

## 6 CONCLUSION

In this paper, we introduce SPACoder, addressing the limitations of existing methods for code translation. By leveraging semi-supervised parallel code generation with enhanced cross-lingual alignment, SPACoder overcomes the challenges of generating high-quality and well-aligned parallel code data in a scalable manner. We demonstrate the effectiveness of SPACoder through extensive experiments conducted on multiple languages and models. The synthetic parallel code generated by SPACoder significantly improves the performance of code translation, outperforming state-of-the-art baselines by a significant margin. Notably, our method achieves remarkable gains in C translations even with a limited number of annotated training instances. Our work showcases the importance of generating parallel code data with good quality and alignment in order to enhance code translation capabilities. Future work can extend to more tasks that benefit from large amount of parallel data.

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
