# SPACODER: ALIGNMENT-ENHANCING PARALLEL CODE GENERATION FOR SEMI-SUPERVISED CODE TRANSLATION

## A  CODE AND DATA SHARING

The code and data for this work are shared through this anonymous link [1].

## B  PARALLEL CODE DATA

Parallel code data refers to code pairs from different programming languages that are functionally equivalent and bug-free. One type of existing datasets is characterized by relatively high alignment but are limited in size and supported languages. For example, CodeXGLUE (Lu et al., 2021) constructed a Java – C# translation dataset by matching function names from open-source repositories. MuST-PT (Zhu et al., 2022) introduced a program translation dataset CoST, with snippet-level alignment that supports 7 programming languages. CoST was collected from coding tutorial website GeeksforGeeks[2], where each coding problem is provided with solutions in up to 7 languages, with each in similar structure and comments. AVATAR (Ahmad et al., 2021b) only supports the translation between Java and Python. Another type of datasets is usually significantly larger in size and supports a wider range of languages, but the alignment quality is low. They are usually collected from competitive online code judgments. Given a coding problem, users can submit their own solutions in various supported languages and get judged based on online tests. The user-contributed solutions to the same problems are collected as parallel code in different languages. For example, Google Code Jam and Project CodeNet (Puri et al., 2021) were both collected in this manner. However, due to the diverse background and the large number of users, the solutions for the same problem have wide discrepancies in distribution across different languages, which lowers the quality of the alignment.

## C  IMPLEMENTATION DETAILS.

All models are trained with a batch size of 16 for 10 epochs, with a learning rate of $5e-5$. Experiments are performed on one NVIDIA A100 GPU with 80G memory. For tempered sampling, we use a sample size of 10 with a fixed temperature of $0.5$. For evaluation, we use beam search with a beam size of $5$. We use a max sequence length of 200 tokens for both the inputs and outputs.

**Preprocessing** For all the program data, we first remove all the comments, docstrings, and empty lines. New lines are replaced with special token NEW_LINE. For pre-tokenization, Python is pre-tokenized with a TreeSitter-based tokenizer from TransCoder(Roziere et al., 2020), for better handling of indentations. Other languages are not pre-tokenized. When running experiments, the data will be tokenized again using the corresponding tokenizer of each model.

**Function Info Extraction.**  We rely on AST parsing to extract function information from programs, which are further used for static analysis and execution-based evaluation. An AST is a tree-like data structure that represents the structure of a program's source code. It captures the high-level structure of the code and the relationships between its elements, enabling a deeper understanding of the code beyond the sequence-level. To create an AST, the source code is first parsed to identify its syntactic elements, such as keywords, operators, and identifiers. The parser then constructs the AST by

---

[1] https://drive.google.com/drive/folders/1hUbfHNMINgrBng1HW8jhsrb4fG5ECT_J?usp=sharing

[2] https://www.geeksforgeeks.org/

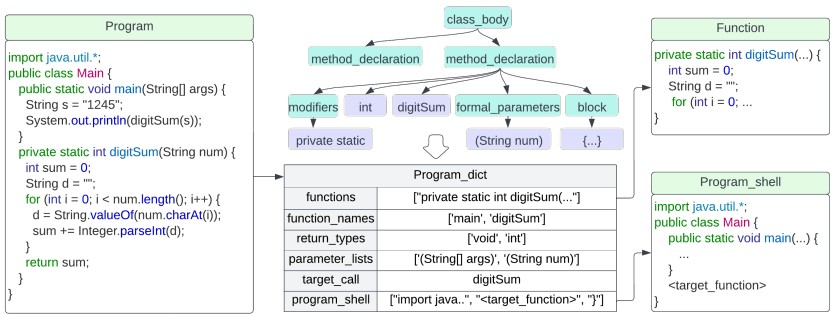

Figure 1: An illustration of function info extraction through AST parsing. Given an input program, we first generate its corresponding AST, and then extract function-related information from AST into program_dict. The tree in the top middle shows an example of AST. After the functions are extracted, the leftover part of the program is called `program_shell`, which can be used for execution-based evaluation later.

assigning each syntactic element to a node in the tree. An AST consists of terminal and non-terminal nodes. Terminal nodes are leaf nodes in AST, and are part of the source code. Non-terminal nodes are not part of the source code. With the help of AST, we can extract function related information by matching the corresponding non-terminal nodes in that language, such as `method_declaration`, `method_invocation`, `formal_parameters` etc. One of the most widely used open-source AST parsing tool is TreeSitter[3]. It supports most of the commonly used programming languages. Figure 1 shows an example of a Java program and its AST (parsed by TreeSitter). The blue nodes are non-terminal and the purple nodes are terminal.

**Sourcing of Monolingual Code Inputs** CODENET (Puri et al., 2021) is a huge dataset containing 13 million of programs in 55 languages. The programs in CODENET are from code submissions to online judge websites of programming problems. We use CODENET as a source of monolingual code inputs for parallel code generation. We select the "Accepted" submissions (submissions that pass the prescribed tests) in 4 languages, C++, Java, Python, and C, from around 1600 problems, which gives us approximately 1 million programs. To ensure the quality of the input data, we set two filtering criteria: (1) the program should be modularized, which means it should contain at least one function (other than `main()` or `Main()` function), and (2) the program should be bug-free, which means it can be compiled without errors. After applying the two steps of filtering, only around $8\%$ of the programs remain, approximately 87k.

**Parallel Code Generation** We experiment with two different models as the generator model, PLBART (Ahmad et al., 2021a) and CodeT5 (Wang et al., 2021). The generator models are initialized by first training on the snippet-level data, and then the function-level data from ECOST. We then utilize the monolingual CODENET data as inputs and acquire the hypotheses from the generators through tempered sampling. For cross-lingual static analysis, we extract the function information of both the monolingual inputs and all the hypotheses and compare them. For compilation, we use the compiler of each language to compile all the hypotheses. Since the hypotheses are functions not programs, we pair each of them with a set of common imports in the corresponding language before compilation to avoid dependency errors. For Python, we first try with python2, and subsequently with python3 if python2 returns with an error. The statistics of the selected hypotheses generated by SPACoder-CodeT5 can be found in Table 3. Table 1 shows all the datasets used for training and how they are acquired.

**Execution-Based Evaluation.** ECOST test set is used for all the evaluations. ECOST train set and generated parallel data are used for model training. The train/valid/test split of ECOST is 70:5:25, and the generated parallel dataset is 85:5:10. The statistics of ECOST is shown in Table 2. To evaluate the quality of the generated hypotheses, we employ an execution-based evaluation strategy. By inserting the generated hypothesis of an input function into the `program_shell` of the ground truth program, we execute the modified program and compare its output against the original output. This process

---

[3]https://tree-sitter.github.io/tree-sitter/

| Data | Type | Volume | Source |
|------|------|--------|--------|
| Function | Annotated | 3326 | ECoST |
| Snippet | Annotated | 31818 | ECoST |
| AND | Synthetic | 69358 | Static Analysis & Compilation |
| COMP | Synthetic | 79550 | Compilation |
| STAT | Synthetic | 176482 | Static Analysis |
| BT | Synthetic | 260110 | Back Translation |

Table 1: Datasets used for the Alignment-Ascending curriculum learning. Volume means number of parallel codes.

| CoST | Function-Level | | | | | | Snippet-Level | | | | | |
|------|--------|--------|-------|--------|--------|------|--------|--------|-------|--------|--------|------|
| | C++-Java | C++-Py | C++-C | Java-Py | Java-C | Py-C | C++-Java | C++-Py | C++-C | Java-Py | Java-C | Py-C |
| Train | 1014 | 947 | 138 | 947 | 146 | 134 | 10472 | 8893 | 1358 | 8716 | 1305 | 1074 |
| Val | 51 | 46 | 14 | 47 | 14 | 14 | 417 | 324 | 78 | 340 | 78 | 69 |
| Test | 372 | 332 | 77 | 331 | 73 | 64 | 2493 | 1991 | 450 | 1964 | 422 | 313 |

Table 2: Data split and number of parallel code pairs in ECoST.

allows us to verify whether the hypothesis successfully passes the built-in test cases, thus evaluating its correctness and suitability. However, the function names in the generated hypotheses might not match the function calls in `program_shell`, causing execution errors. Therefore, through function information extraction, we replace the function name of the hypotheses with the corresponding ground truth function name before each evaluation.

# D   EFFECT OF REARRANGING THE CURRICULUM.

In Table 4, we presented some new experimental results (with PLBART). Here is an overview of each experiment:

- Function: trained on ECoST-function (annotated data).
- Snippet+Function: trained on ECoST-function and ECoST-snippet (annotated data with fine-grained alignment).
- AND+Snippet+Function: trained with AND (high quality synthetic data filtered by both static analysis and compilation).
- BT+Snippet+Function: trained with BT (noisy synthetic data with no filtering).
- AND+COMP+STAT+BT+Snippet+Function: SPACoder with reversed curriculum order on synthetic data.
- BT+STAT+COMP+AND+Snippet+Function: SPACoder

We can see that when using the high quality synthetic data, AND, the performance is not as good as SPACoder, which uses larger amounts of noisier synthetic data. Similarly, when using the unfiltered noisy data, BT, the performance is also subpar. This shows that both the quality and the quantity of the synthetic data hold significant importance to the performance. When we reverse the order of the curriculum to AND+COMP+STAT+BT+Snippet+Function, the performance drops significantly compared to SPACoder, which indicates the importance of the order of the curriculum to the performance. It is worth noting that the reversed curriculum has very similar performance with

| CodeT5 | Number of Pairs | | | | | | Selection Rate | | | | | |
|--------|--------|--------|-------|--------|--------|------|--------|--------|-------|--------|--------|------|
| Selector | C++-Java | C++-Py | C++-C | Java-Py | Java-C | Py-C | C++-Java | C++-Py | C++-C | Java-Py | Java-C | Py-C |
| Back Translation (BT) | 47637 | 64037 | 49550 | 37422 | 22935 | 39335 | 1 | 1 | 1 | 1 | 1 | 1 |
| Static Analysis (STAT) | 25211 | 58663 | 14945 | 31379 | 13059 | 34072 | 0.53 | 0.92 | 0.30 | 0.84 | 0.57 | 0.87 |
| Compilation (COMP) | 17373 | 36544 | 2290 | 16888 | 3821 | 13947 | 0.36 | 0.57 | 0.05 | 0.45 | 0.17 | 0.35 |
| SA & Compilation (AND) | 10811 | 35457 | 1325 | 15256 | 2731 | 13309 | 0.23 | 0.55 | 0.03 | 0.41 | 0.12 | 0.34 |

Table 3: Statistics of CODENET-SPACoder, with CodeT5 (Wang et al., 2021) as generator. SA & Compilation refers to the intersection of the Static Analysis and Compilation selections.

| | | Computation Accuracy | | | | | |
|---|---|---|---|---|---|---|---|
| Model | Data Volume | Java-C++ | Py-C++ | C++-Java | Py-Java | C++-Py | Java-Py |
| Function | 3326 | 0.81 | 4.52 | 1.88 | 3.63 | 16.87 | 16.62 |
| Snippet+Function | 35144 | 25.54 | 24.4 | 27.15 | 23.87 | 32.23 | 32.33 |
| AND+Snippet+Function | 104502 | 34.68 | 34.64 | 33.06 | 32.93 | 36.45 | 37.16 |
| BT+Snippet+Function | 295254 | 38.98 | 34.94 | 37.1 | 30.21 | 35.54 | 39.58 |
| AND+COMP+STAT+BT+Snippet+Function | 551286 | 38.98 | 32.23 | 37.63 | 33.84 | 35.84 | 39.58 |
| BT+STAT+COMP+AND+Snippet+Function (SPACoder) | 551286 | 41.94 | 35.24 | 40.05 | 33.84 | 38.55 | 41.09 |

Table 4: Comparison with variations of curriculums. The model used is PLBART. Data Volume means number of parallel codes.

BT+Snippet+Function, likely due to the larger volume of the BT dataset overpowering the effect of the previous datasets.

## E  LIMITATIONS AND FUTURE WORK

Despite the promising results and contributions of our work, there are several limitations that need future research. Our method relies heavily on the generation of parallel code data and does not take into account other types of information that may be useful for code translation, such as comments or documentation. Incorporating such information into the generation process could potentially further improve the quality of the generated data. Additionally, our evaluation is mainly focused on execution-based metrics, which measure the quality of the generated code based on its ability to execute correctly. While these metrics are important, they do not capture other aspects of code quality, such as readability, maintainability, or style. Future work could explore the development of metrics that capture these aspects of code quality.

## F  BROADER IMPACT

The ability to automatically translate code between programming languages can help software developers to port existing codebases from one language to another, allowing them to work with a wider range of tools and frameworks. It can also facilitate collaboration between developers who work with different programming languages. In addition, our work has the potential to reduce the barriers to entry for new developers who want to learn a new programming language. By enabling them to translate code from a language they are familiar with to a new language, they can quickly learn the connections and differences between the two languages, and start working on projects in the new language. Moreover, it also has the potential to create more inclusive software engineering learning environments, which makes computer science more accessible for learners from various backgrounds. However, there are also potential negative impacts of this work, such as the possibility of automated code translation leading to loss of jobs for software developers or increased reliance on automated tools in the software development process.