# OpenReview forum: "Alignment-Enhancing Parallel Code Generation for Semi-Supervised Code Translation"
_ICLR.cc/2024/Conference — Submitted to ICLR 2024_

### Official Review · Reviewer_okeA · 2023-10-22

**Soundness:** 3 good
**Presentation:** 3 good
**Contribution:** 2 fair
**Rating:** 5
**Confidence:** 5

**Summary:**

They propose a semi-supervised code translation method, SPACoder, that leverages snippet training, static analysis, and compilation to generate synthetic parallel code with enhanced alignment in a scalable way, and improves code translation by curriculum learning based on the alignment level of training instances. SPACoder can be generalized to multiple languages and various models with little overhead.

**Strengths:**

- The curriculum learning improves the generation's performance
- They propose some methods for generating synthetic codes

**Weaknesses:**

- The novelty of this paper is limited. The synthetic generation and alignment-ascending curriculum learning seems simple and straightforward.
- The discussion of the baseline is not enough. I mean, I cannot get what the contribution this paper achieved.
- during the generation of synthetic code, can you generate the snippet-level alignment?

**Questions:**

- See above.
- During selecting the synthetic code, why not run the code and compare the returned results directly？

---

> ### Author Response · Authors · 2023-11-20
> **Thank you for your valuable feedback**
>
> Weaknesses:
> - The novelty of this paper is limited. The synthetic generation and alignment-ascending curriculum learning seems simple and straightforward.
>
> **Response:** Thank you for acknowledging that our method is simple and straightforward. We believe that when combined with effectiveness in improving the performance, simplicity and straightforwardness can be strengths instead of weaknesses.
>
> - The discussion of the baseline is not enough. I mean, I cannot get what the contribution this paper achieved.
>
> **Response:** Thank you for your feedback. We discussed some of the baselines in the related work section, hence we curated the detailed introduction to each of them in the experiment section. We are ready to answer any specific questions about the baselines if you can kindly share them.
>
> - During the generation of synthetic code, can you generate the snippet-level alignment?
>
> **Response:** The snippet-level aligned data is from an existing dataset, CoST (Zhu et al.). We did not generate them during the parallel code generation.
>
> - During selecting the synthetic code, why not run the code and compare the returned results directly？
>
> **Response:** The synthetic code is function-level code, and to run them requires necessary dependencies and testcases. It means that we need customized cross-lingual testcase generation for every single code in every language, which is challenging and expensive to acquire. For example, to test 260110 codes generated by BT in table 7 (shown below), at least 260110 different test cases need to be generated in 4 different languages. On the other hand, compilation filtering only takes the compiler of the target language, and neither compilation nor static analysis require customization for each code. Therefore, our methods is significantly cheaper and can be extended to more languages with little overhead.
>
> **Table 7: Datasets used for training. Volume means number of parallel codes.**
>
> | Data     | Type      |  Volume | Source                        |
> |----------|-----------|---------|-------------------------------|
> | Function | Annotated |    3326 | ECoST                         |
> | Snippet  | Annotated |   31818 | ECoST                         |
> | AND      | Synthetic |   69358 | Static Analysis & Compilation |
> | COMP     | Synthetic |   79550 | Compilation                   |
> | STAT     | Synthetic |  176482 | Static Analysis               |
> | BT       | Synthetic |  260110 | Back Translation              |

---

> > ### Comment · Reviewer_okeA · 2023-11-21
> > **Change my rating**
> >
> > Thanks for your rebuttal and some of my concerns have been solved.  I will raise my rating. However, I still believe the novelty and contribution of this paper do not reach the acceptance criteria of ICLR.

---

> > > ### Author Response · Authors · 2023-11-22
> > > **Thank you for your valuable feedback**
> > >
> > > Dear reviewer okeA,
> > >
> > > Thank you for taking the time to revisit our paper and for acknowledging the efforts made in our rebuttal. The positive comments provided by reviewers highlight the novel aspects, strong empirical performance, and impact of our work. We value your feedback and appreciate your consideration and the adjustment made to your rating. Thank you for your thoughtful consideration of our work.

---

### Official Review · Reviewer_jLZb · 2023-10-31

**Soundness:** 4 excellent
**Presentation:** 2 fair
**Contribution:** 3 good
**Rating:** 6
**Confidence:** 4

**Summary:**

Current neural code translation approaches are of two kinds.

1. Those that rely on unsupervised “back translation” and/or denoising auto-encoding. These methods do not learn alignment between languages in a supervised fashion, and sometimes produce low-quality translations that translate token-by-token with incomplete understanding of target language semantics (the authors of this paper refer to this as “shallow translation”).

2. Those that rely on supervised fine-tuning on parallel aligned data. The problem here is that high quality parallel code data is very hard to come by.

This paper works with the second family of approaches, and attempts to solve the insufficient data problem by proposing a method to generate high quality parallel aligned data for supervised fine-tuning.

The method is conceptually simple: take a small quantity of existing parallel data and train a model $f_G$ on it. Then take a large amount of monolingual data and pass each function through the model $f_G$. Filter out incorrect translations by matching function arguments and return types (a static analysis filter), and checking if it compiles (compilation filter).

Instead of applying all the filters together, the authors propose a curriculum learning framework, where they first train the model on the *unfiltered* data, and train the model on progressively more and more filtered data.

The evaluation is of two kinds - a) they show that a model trained on their parallel data performs better than the **same** model trained on parallel data from other sources, b) they show that a model trained on their parallel data with a curriculum learning approach performs better than **other** existing models.

**Strengths:**

1. Conceptually simple approach.
1. Very relevant problem with good impact. It's true that back-translation suffers from "shallow translation", and supervised fine-tuning suffers from data scarcity.
1. It is clear from the evaluation that the synthetic parallel data is of good quality, and there is a clear benefit to using this data for fine-tuning for the downstream task of translation.

**Weaknesses:**

I think overall this paper is a good contribution to the field and should be published. However, I think there are several places where there is a lack of precision and clarity in the writing/terminology. There are also some non-intuitive concepts / details that are skimmed over, and could benefit from some elaboration. The paper would be much easier to read if these were fixed. These are listed in the Questions section.

**Questions:**

**Unclear/Imprecise terminology:**

1. Section 2 - “the paper focuses on function-level code translation”. This confused me because the final evaluation is on computational accuracy, which cannot be evaluated at the function level. I think you mean to say that the parallel aligned data is generated at a function level, but the technique is evaluated on full source files?

1. The word “snippet” is vague and is used to mean different things in different places. For example, in Section 2.1 - “...takes as input a code snippet $x$” - here, snippet means *function*, presumably. But then in Section 2.1.1, there is a contrast between “snippet level” and “function level”, suggesting that snippets are smaller than functions. Could you please define what a snippet is, somewhere early on?

1. What is “SPACoder-function” (Table 2 and Section 4.1)? Is it BT, STAT, COMP or AND? Or is it *all* of them, but in a curriculum learning setup?

1. When you use the terminology “SPACoder-PLBART” or “SPACoder-T5”, there are actually two models involved here - the original *generator* model used to produce the synthetic parallel data, and the *base model* that you’re fine-tuning. I assume “SPACoder-PLBart” refers to a model where the generator *as well as* the base model are PLBART? Would be nice if this was clarified.

**Concepts that need more elaboration:**

1. When doing static analysis on function signatures, how do you match types between different languages? Like int[] in Java and vector< int > in C.

1. If you are operating at the function level, how do you apply a compilation filter? In C/C++, you can compile individual functions without linking, but not in Java. And Python code is not compiled, just interpreted. It would be nice if this was clarified.

1.  Let us say that the generator takes code from Language A and converts it to Language B. Then while fine-tuning, do you train on [B, A] samples, or [A, B] samples or both? In other words, while fine-tuning, which is the source language and which is the target language? This is also related to my next comment below.

1. Section 2.2 - “Without the selector, the generation is reduced to plain back-translation”. I’m having difficulty understanding why this is true. According to me, this would only be true if 1) the generator takes code from Language A and converts it to Language B, but you fine-tune another model on [B, A] samples. 2) both the generator and the fine-tuning model are *trained together* (back translation relies on this kind of joint improvement of the forward and the backward model). Could you please clarify this terminology?

1. Typically, curriculum learning starts with **easy** examples and moves to **difficult** examples. Here, it seems like it starts with **low-quality** examples and moves to **high-quality** examples, which is non-intuitive ([low-quality ~ easy] and [high-quality ~ difficult]?). After a little thought, I think I understand why this is set up like this, but it is an important subtlety and should be clarified.

**Typos / Bad phrasing:**

Section 2.2 - “we denote the synthetic parallel data from cross-lingual static analysis as STAT and COMP, respectively” - this line needs fixing.

---

> ### Author Response · Authors · 2023-11-19
> **Thank you for your valuable feedback**
>
> Questions:
>
> - Unclear/Imprecise terminology: function-level code translation.
>
> **Response**: Your understanding is correct. The parallel code generation is at function level, and the computational accuracy evaluation is done by wrapping the functions into executable programs, which contains the necessary dependencies and test cases.
>
>
> - The word “snippet” is vague.
>
> **Response**:  The word “snippet” in this paper means small chunks of code that are separated by code comments in a program. It is not necessarily a complete function. We will add the definition of both early in the draft.
>
>
> - What is “SPACoder-function” (Table 2 and Section 4.1)? Is it BT, STAT, COMP or AND? Or is it all of them, but in a curriculum learning setup?
>
> **Response**:  SPACoder-function is a dataset that contains all of BT, STAT, COMP and AND in a curriculum learning setup. Compared to the final dataset we used to train the SPACoder models, the only difference is that it does not include the annotated data ECoST-*.
>
> - I assume “SPACoder-PLBart” refers to a model where the generator as well as the base model are PLBART?
>
> **Response**:  Your understanding is correct. SPACoder-PLBART means a PLBART model trained on PLBART generated data, same as SPACoder-CodeT5. This setup is to demonstrate that our proposed method can be generalized to different models, without the assistance of more powerful models.
>
> Concepts that need more elaboration:
>
> - When doing static analysis on function signatures, how do you match types between different languages? Like int[] in Java and vector< int > in C.
>
> **Response**: We build the cross-lingual type matching rules by extracting function information from large code corpora like CodeNet through AST parsing. After AST parsing, the information about a function will be stored in a dictionary, from which we can access each function’s signature. All the types will first be converted to a small set of base types, and then checked for more fine-grained alignment. For example, ArrayList in Java and <vector> in C++ will both be mapped to the base type list. The Integer type in Java will be converted to base type int, which can be mapped to the int type in C++. In this way, ArrayList<Integer> and vector< int > will be labeled as a cross-lingual pair of matching types.
>
> - If you are operating at the function level, how do you apply a compilation filter?
>
> **Response**: The compilation is done by creating a temporary file with the given code and calling the corresponding compiler from the command line. For Java, the function will be wrapped in a class with an empty main() function, and run with the javac command. For Python, the file will be run with the “python2/python3” command, which is the same as executing the code. After each compilation, the temporary file is deleted and the stdout/stderr will be saved for filtering and evaluation later. This whole process is parallelized, with memory limit and timeout for each process.
>
> - Let us say that the generator takes code from Language A and converts it to Language B. Then while fine-tuning, do you train on [B, A] samples, or [A, B] samples or both?
>
> **Response**:  We use both. We have conducted experiments to compare the performance of using A-B, B-A and both as training data, and found that using both consistently gives better performance. Therefore we have been using the combination of both A-B and B-A for all the experiments.
>
> - Could you please clarify this terminology back-translation?
>
> **Response**:  Your understanding is correct. Since we use data generated in both directions, it is indeed different from commonly known back-translation. What we refer to is “parallel data generation with trained translator models”, and for simplicity, we call it back-translation since it’s conceptually similar. We will add the clarification in the draft.
>
> - Typically, curriculum learning starts with easy examples and moves to difficult examples.
>
> **Response**: Thank you for your feedback. You are right that the alignment-ascending curriculum in this paper appears slightly different from the typical easy-to-difficult curriculum. However, just like you kindly already pointed out, in terms of programming languages, high-quality data can actually be more challenging than low-quality data. Because low-quality parallel data might share more similarities between the source and target language due to the “shallow translation” problem, where the model translates token by token or directly copies from the source language. This makes the low-quality data potentially easier to learn than the high-quality data. We agree that it is important to clarify the subtlety and we will add it in the draft.
>
> Typos / Bad phrasing:
>
> **Response**: Thank you for your feedback. We will rephrase it to “we denote the synthetic parallel data from cross-lingual static analysis and compilation filtering as STAT and COMP, respectively”.

---

### Official Review · Reviewer_9b25 · 2023-11-01

**Soundness:** 2 fair
**Presentation:** 3 good
**Contribution:** 2 fair
**Rating:** 6
**Confidence:** 5

**Summary:**

This paper introduces a method called SPACoder for improving the translation of source code from one programming language to another. The paper argues that one of the main challenges in training neural code translation models is the limited availability of parallel code data in different languages. SPACoder addresses this issue through a semi-supervised approach that first prompts the pre-trained models to translate code from a language to another and then select the better aligned snippets to further train the model. The paper proposed to apply static analysis, and compilation to select the synthetic parallel code examples with better alignment. It also employs curriculum learning based on alignment levels to enhance code translation. SPACoder is versatile, applicable to multiple programming languages and various models with minimal additional overhead. Experimental results demonstrate that SPACoder significantly enhances code translation performance in languages like C++, Java, Python, and C, outperforming state-of-the-art methods, even with a small number of annotated training instances, such as improving C translation by up to 43%.

**Strengths:**

- The overall workflow of SPACoder is intuitive and straightforward to implement yet achieves improvements over existing approaches. It proposes to use static analysis and compilation to estimate the alignment of the parallel data, which alleviates the burden of the selection process that heavily relies on execution correctness.

- The application of curriculum learning in translation is reasonable since the direct learning of the alignment among several different programming languages is difficult for the model to learn and generalize.

- I like the overall presentation. For instance, the examples in Figures 1, 3, and 4 clearly demonstrate the weaknesses of the previous code translation model and the improved performance of SPACoder. The related work section is also very well-written.

**Weaknesses:**

- The comparison to TransCoder-ST is not well controlled, and the explanation regarding the comparison results requires further explanation. To me, the most relevant baseline to SPACoder is TransCoder-ST, where both share the high-level idea of firstly generating the translation by the model itself, then selecting better-aligned data with some estimation and finally reinforcing the model’s prediction towards these better-aligned samples while avoiding those misaligned. The main novelty of SPACoder lies in (1) it proposes to use the light-weight static analysis to replace the dynamic correctness as the selection strategy, (2) it proposes to eventually increase the learning difficulty for the model. However, there are the following issues when comparing to Transcoder-ST.
   1. First, a strictly controlled comparison is missing where SPACoder should be, similar to TransCoder-ST, initialized from the vanilla TransCoder, and such a SPACoder-TransCoder could isolate the comparison between static analysis + curriculum learning vs. dynamic analysis.

  2. It is not clear why SPACoder-PLBART keeps loosing to TransCoder-ST in Computation Accuracy, and it seems the effectiveness of SPACoder largely depends on the quality of the pre-trained checkpoints. However, it is strange that the vanilla PLBART is comparable or better than CodeT5 across Py2Java, CPP2Py, Java2Py, in computation accuracy, and Py2CPP, Py2Java, CPP2Py in CodeBLEU, while such trends are completely reversed when the model is further trained with SPACoder strategy. I would urge the authors to analyze the weaknesses of SPACoder-PLBART and explain the reversed trends in detail.

3. The ablation study doesn’t support the effectiveness of curriculum learning:
   - The result for BT + STAT + COMP + AND is missing.
   - To show the effectiveness of curriculum learning, results for rearranging the training stages should be shown.
   - It’d be even better to show the result of training only on the AND dataset for the same amount of total computation as curriculum learning and compare the two results.

- Given the unstable performance of the SPACoder I mentioned above, I would like to see more results using larger models. As the inconsistent improvement SPACoder brought to PLBART and CodeT5, I would encourage the authors to extend their variant set to more models. Besides the TransCoder version I mentioned above, the codeT5-large, and the codet5+ family of varied decoder sizes might be worth trying to illustrate the generalizability and the effectiveness of SPACoder.

- It is not clear why compilation mostly hurts the performance. In the ablation study of Table-5, it is not explained why compilation mostly hurts the model’s performance during the curriculum learning. This is a bit counterintuitive, since compilation should be able to help filtering out those useless pairs and removing them could make the models focus on predicting at least compiled code. This downgrade of performance requires further analysis.

- It’s not clear how much more total computation that SPAcoder takes when compared to previous models. Considering the curriculum learning during training, SPAcoder might have been trained on the finetuning dataset for more epochs when compared to previous models.

**Questions:**

- It’d be great to see a comparison of the total computation in the main results.
- It’d be great to see a thorough ablation study of the effectiveness of curriculum learning.
- It’d be great to see experiments demonstrating the efficiency/scalability of Static Analysis filtering compared to Test Case filtering.
- What is the version of CodeT5? Small? Base? Please specify.

---

> ### Author Response · Authors · 2023-11-19
> **Thank you for your valuable feedback**
>
> **Table 6: Model size and number of parameters.**
>
> | Model         | Size | Parameters |
> |---------------|------|------------|
> | CodeBERT      | base | 125M       |
> | PLBART        | base | 139M       |
> | CodeT5        | base | 220M       |
> | TransCoder    | -    | 110M       |
> | TransCoder-ST | -    | 110M       |
>
>
> **Table 7: Datasets used for training. Volume means number of parallel codes.**
>
> | Data     | Type      |  Volume | Source                        |
> |----------|-----------|---------|-------------------------------|
> | Function | Annotated |    3326 | ECoST                         |
> | Snippet  | Annotated |   31818 | ECoST                         |
> | AND      | Synthetic |   69358 | Static Analysis & Compilation |
> | COMP     | Synthetic |   79550 | Compilation                   |
> | STAT     | Synthetic |  176482 | Static Analysis               |
> | BT       | Synthetic |  260110 | Back Translation              |
>
> **Table 8: Comparison with new baselines (variations of curriculums). Data Volume means number of parallel codes. The evaluation metric is Computation Accuracy.**
>
> | Model                                        | Data Volume | Java-C++ | Py-C++ | C++-Java | Py-Java | C++-Py | Java-Py |
> |----------------------------------------------|-------------|----------|--------|----------|---------|--------|---------|
> | function                                     |        3326 |     0.81 |   4.52 |     1.88 |    3.63 |  16.87 |   16.62 |
> | snippet+function                             |       35144 |    25.54 |   24.4 |    27.15 |   23.87 |  32.23 |   32.33 |
> | AND+snippet+function                         |      104502 |    34.68 |  34.64 |    33.06 |   32.93 |  36.45 |   37.16 |
> | BT+snippet+function                          |      295254 |    38.98 |  34.94 |     37.1 |   30.21 |  35.54 |   39.58 |
> | AND+COMP+STAT+BT+snippet+function            |      551286 |    38.98 |  32.23 |    37.63 |   33.84 |  35.84 |   39.58 |
> | BT+STAT+COMP+AND+snippet+function (SPACoder) |      551286 |    41.94 |  35.24 |    40.05 |   33.84 |  38.55 |   41.09 |
>
> **Model size and compute:**
> In Table 6 (shown above), we presented the size and number of parameters of each model.
>
> **Comparison between testcase filtering and compilation/static analysis filtering:**
> 1. Difference in cost. Testcase filtering requires customized cross-lingual testcase generation for every single code in every language, which is challenging and expensive to acquire. For example, to test 260110 codes generated by BT in table 7, at least 260110 different test cases need to be generated in 4 different languages. On the other hand, compilation filtering only takes the compiler of the target language, and neither compilation nor static analysis require customization for each code. Therefore, our methods is significantly cheaper and can be extended to more languages with little overhead.
> 2. Difference in hypothesis. Testcase filtering is essentially an alternative way to acquire annotated parallel code data, because target code that passes expected testcases is functionally parallel to the source code. Therefore, testcase filtering follows the straightforward hypothesis of more annotated data leads to better code translation performance. On the other hand, our hypothesis is that large amounts of noisy parallel data arranged in a specific curriculum benefits code translation more than small amounts of annotated data. Our experimental results (with more presented in Table 8) support this hypothesis.
>
>
> **Effect of rearranging the curriculum:**
>
> In Table 8 (shown above) , we presented some new experimental results (with PLBART). Here is an overview of each experiment:
> - Function: trained on ECoST-function (annotated data).
> - Snippet+Function: trained on ECoST-function and ECoST-snippet (annotated data with fine-grained alignment).
> - AND+Snippet+Function: trained with AND (high quality synthetic data filtered by both static analysis and compilation).
> - BT+Snippet+Function: trained with BT (noisy synthetic data with no filtering).
> - AND+COMP+STAT+BT+Snippet+Function: SPACoder with reversed curriculum order on synthetic data.
> - BT+STAT+COMP+AND+Snippet+Function: SPACoder
>
> We can see that when using the high quality synthetic data, AND, the performance is not as good as SPACoder, which uses larger amounts of noisier synthetic data. Similarly, when using the unfiltered noisy data, BT, the performance is also subpar. This shows that both the quality and the quantity of the synthetic data hold significant importance to the performance. When we reverse the order of the curriculum to AND+COMP+STAT+BT+Snippet+Function, the performance drops significantly compared to SPACoder, which indicates the importance of the order of the curriculum to the performance. It is worth noting that the reversed curriculum has very similar performance with BT+Snippet+Function, likely due to the larger volume of the BT dataset overpowering the effect of the previous datasets.

---

> ### Author Response · Authors · 2023-11-22
> **Thank you for your valuable feedback**
>
> Dear Reviewer 9b25, in light of the updates made in the rebuttal, we would be grateful if you could comment on whether our response addressed your concerns. Thank you for your time and consideration.

---

> > ### Comment · Reviewer_9b25 · 2023-11-22
> >
> > Thanks for the detailed answer. While most of my answers are addressed, SPACoder essentially increases the volume and computation of training significantly. This needs to be acknowledged.
> >
> > It is still not clear to me why the performance trend varies CodeT5 and PLBART.
> >
> > Nonetheless, I like the work and increase my score.

---

> > > ### Author Response · Authors · 2023-11-23
> > >
> > > Thank you very much for re-evaluating our work and increasing the score. We greatly appreciate your recognition of our efforts in addressing your concerns in the rebuttal.
> > >
> > > We acknowledge the importance of transparency about the computational costs involved and will further revise the paper to include this information. While SPACoder leverages large amounts of unlabelled data for semi-supervised learning and consequently requires more compute compared to supervised learning on small annotated datasets, it is important to consider its relative cost-efficiency compared to other semi-supervised/unsupervised learning methods. For example, static analysis and compilation used in our method are inherently less computationally demanding than test case execution used in TransCoder-ST.
> > >
> > > As for the varying performance trends between CodeT5 and PLBART, we acknowledge that this aspect of our results could benefit from further investigation. Due to CodeT5's size being significantly larger than PLBART's, one hypothesis could be that model size is related to how much the model benefits from large amounts of noisy data compared to small amounts of annotated data. We plan to delve deeper into this in our future work.
> > >
> > > Once again, we are thankful for your constructive feedback and for the increased score. Your comments have been instrumental in refining our paper and guiding our future research directions.

---

### Official Review · Reviewer_zW1g · 2023-11-01

**Soundness:** 3 good
**Presentation:** 4 excellent
**Contribution:** 3 good
**Rating:** 6
**Confidence:** 4

**Summary:**

This paper presents a technique for preparing synthetic parallel data to train code translation model. The key idea is to leverage sampling of parallel data from a base model, but then leverages AST analysis and compilation check to filter out low quality data to produce higher quality synthetic data to help with train the model.

The paper takes advantages of curriculum training, starting from snippet level alignments then to function alignment data. The evaluation shows that the SPACoder improves performance of both PLBART and CodeT5, and curriculum helps improvements of the overall model performance.

**Strengths:**

1. The paper's main contribution is the idea of using AST similarity to filter synthetic data to improve its quality for code translation task.
2. The use of curriculum learning to help bootstrap the training
3. Comprehensive experiments comparing against both zero-shot and finetuned baselines.

I believe the idea of leveraging AST similarity is quite novel for the given task --- at least for the languages considered, their AST structural similarity is indeed a signal could help with improving dataset quality. While I doubt this technique would be directly available for training true low-resource languages (e.g., translation from Java to DSLs like Halide) given their AST difference, I think this technique could still inspire researchers to consider invariants on AST, or even on control flow graph level that can be used to enhance data similarity. Given that this paper did a good job finding such AST invariants and engineering it to solve code translation task, I think this paper deserves attention from the community.

**Weaknesses:**

The paper lacks some comparison with newer public models (StarCoder, CodeLLama etc), or maybe closed source commercial model like GPT-3.5. While such comparisons may seem like "comparing apple to pear" due to their differences in model size and corpus, I believe they are necessary if the authors want to show that SPACoder is truly advancing the problem of code translation. For larger langauge models, they often make much less compilation or runtime errors, and many of the problems appear in smaller models like CodeT5 would disappear. If that's the case, the improvement using AST augmentation would be smaller, given that their main goal is to reduce syntax and simple run-time errors.

The authors argue the effectiveness of the technique on "low-resource" language C. While this is true for the given dataset that parallel C data is much smaller, the community won't agree C is a true low-resource language given that C has the largest size in many public pretraining dataset (e.g., the Stack). If the author truly wants to demonstrate the performance of SPACoder on a resource language, some DSL would be a good target.

The paper also missed an experiment to compare no-curriculum vs curriculum training in terms of BT -> STAT -> COMP -> AND. What would the model performance would be like if you directly finetune PLBart or CodeT5 on AND data without other steps? This would explain whether curriculum  or the dataset matter more.

**Questions:**

I would like authors answer questions related to comparison with LLMs with zero-shot or few-shot experiments, and explain how non-curriculum training would affect the result.

---

> ### Author Response · Authors · 2023-11-20
> **Thank you for your valuable feedback**
>
> **Table 6: Model size and number of parameters.**
>
> | Model         | Size | Parameters |
> |---------------|------|------------|
> | CodeBERT      | base | 125M       |
> | PLBART        | base | 139M       |
> | CodeT5        | base | 220M       |
> | TransCoder    | -    | 110M       |
> | TransCoder-ST | -    | 110M       |
>
>
> **Table 8: Comparison with new baselines (variations of curriculums). Data Volume means number of parallel codes. The evaluation metric is Computation Accuracy.**
>
> | Model                                        | Data Volume | Java-C++ | Py-C++ | C++-Java | Py-Java | C++-Py | Java-Py |
> |----------------------------------------------|-------------|----------|--------|----------|---------|--------|---------|
> | function                                     |        3326 |     0.81 |   4.52 |     1.88 |    3.63 |  16.87 |   16.62 |
> | snippet+function                             |       35144 |    25.54 |   24.4 |    27.15 |   23.87 |  32.23 |   32.33 |
> | AND+snippet+function                         |      104502 |    34.68 |  34.64 |    33.06 |   32.93 |  36.45 |   37.16 |
> | BT+snippet+function                          |      295254 |    38.98 |  34.94 |     37.1 |   30.21 |  35.54 |   39.58 |
> | AND+COMP+STAT+BT+snippet+function            |      551286 |    38.98 |  32.23 |    37.63 |   33.84 |  35.84 |   39.58 |
> | BT+STAT+COMP+AND+snippet+function (SPACoder) |      551286 |    41.94 |  35.24 |    40.05 |   33.84 |  38.55 |   41.09 |
>
> **Comparison with code LLMs:**
>
> In Table 6 (shown above), we presented the size and number of parameters of each model. As shown in the table, all the models are in ther hundred million parameters range. On the other hand, StarCoder has 15.5 billion parameters, and CodeLLama models range from 7 billion to 34 billion parameters. As you have kindly pointed out, comparing our models with code LLMs would be like "comparing apple to pear" due to the significant difference in model size and computation scales.
>
> **Low-resource language:**
>
> We agree that C is not necessarily a low-resource language in the software engineering community. In this paper, we refer to it as a low-resource language because of its significantly smaller training set size (~150 training samples) compared to other languages. We aim to evaluate how our method performs with a weak generator resulted from limited annotated data, and this setting can be extended to more languages with limited training samples. As shown in Table 4 (in the paper), our method improves computational accuracy of C by up to 43%, demonstrating strong performance under the low-resource setting.
>
> **Effect of rearranging the curriculum:**
>
> In Table 8 (shown above) , we presented some new experimental results (with PLBART). Here is an overview of each experiment:
> - Function: trained on ECoST-function (annotated data).
> - Snippet+Function: trained on ECoST-function and ECoST-snippet (annotated data with fine-grained alignment).
> - AND+Snippet+Function: trained with AND (high quality synthetic data filtered by both static analysis and compilation).
> - BT+Snippet+Function: trained with BT (noisy synthetic data with no filtering).
> - AND+COMP+STAT+BT+Snippet+Function: SPACoder with reversed curriculum order on synthetic data.
> - BT+STAT+COMP+AND+Snippet+Function: SPACoder
>
> We can see that when using the high quality synthetic data, AND, the performance is not as good as SPACoder, which uses larger amounts of noisier synthetic data. Similarly, when using the unfiltered noisy data, BT, the performance is also subpar. This shows that both the quality and the quantity of the synthetic data hold significant importance to the performance. When we reverse the order of the curriculum to AND+COMP+STAT+BT+Snippet+Function, the performance drops significantly compared to SPACoder, which indicates the importance of the order of the curriculum to the performance. It is worth noting that the reversed curriculum has very similar performance with BT+Snippet+Function, likely due to the larger volume of the BT dataset overpowering the effect of the previous datasets.

---

> > ### Comment · Reviewer_zW1g · 2023-11-21
> >
> > Thanks for the response. The curriculum ablation results are quite interesting, STAT+COMP+AND seems to play a subtle role on top of BT to produce some improvement.
> >
> > Despite such code translation models are a little overshadowed by more general purpose code LMs nowadays, I remain quite positive about the paper.

---

> > > ### Author Response · Authors · 2023-11-22
> > > **Thank you**
> > >
> > > Dear zW1g reviewer,
> > >
> > > Thank you for your prompt response and positive feedback, we appreciate your recognition of the merit of our work.
> > > Once again, thank you for your time in drafting a constructive review.

---

### Author Response · Authors · 2023-11-21
**Thank all the reviewers for your thorough reviews and insightful suggestions!**

Dear Reviewers,

We would like to express our deepest gratitude for your insightful and constructive feedback on our paper. Your comments have been invaluable in enhancing the quality and clarity of our work. We have revised our draft and supplemental materials to incorporate your suggestions and include new experimental results. We remain open to any further suggestions you may have and are committed to ongoing improvements in our work. Once again, we thank you for your thorough reviews and insightful suggestions.

We are greatly encouraged by your positive comments:

**Novelty:**
- (zW1g) the idea of leveraging AST similarity is quite novel for the given task... Given that this paper did a good job finding such AST invariants and engineering it to solve code translation task, I think this paper deserves attention from the community.
- (9b25) It proposes to use static analysis and compilation to estimate the alignment of the parallel data, which alleviates the burden of the selection process.
- (jLZb) overall this paper is a good contribution to the field and should be published.

**Clarity:**
- (9b25) The overall workflow of SPACoder is intuitive and straightforward to implement yet achieves improvements over existing approaches.
- (okeA) The synthetic generation and alignment-ascending curriculum learning seems simple and straightforward.
- (jLZb) Conceptually simple approach.

**Strong Empirical Performance:**
- (jLZb) It is clear from the evaluation that the synthetic parallel data is of good quality, and there is a clear benefit to using this data for fine-tuning for the downstream task of translation.
- (9b25) The overall workflow of SPACoder is intuitive and straightforward to implement yet achieves improvements over existing approaches.
- (zW1g) Comprehensive experiments comparing against both zero-shot and finetuned baselines.
- (okeA) The curriculum learning improves the generation's performance.

**Presentation:**
- (9b25) I like the overall presentation... The related work section is also very well-written.

**Impact:**
- (jLZb) Very relevant problem with good impact.

---

### Meta-Review · Area_Chair_xP43 · 2023-12-08

**Metareview:**

The paper is clear and technically sound, the authors took an active part in the rebuttal, but the paper falls short of the high bar of publication at ICLR. The impact and novelty is relatively limited, as noted by several reviewers. The method lacks anchor points: the comparison to Transcoder-ST is limited and not particularly convincing, there could be other baselines (e.g. CodeT5+ vs. PLBART), there is no comparison nor discussion of Transcoder-IR [1] (only cited in the intro), which has a related data augmentation approach. Overall, there is not enough experimental evidence to convince the reviewers of the clear usefulness of the approach.

[1] Szafraniec, Marc, et al. "Code translation with compiler representations." arXiv preprint arXiv:2207.03578 (2022).

**Justification For Why Not Higher Score:**

Good ideas, OK execution, but a bit convoluted, and more importantly not really better than a paper from 2021 (Transcoder-ST, Rozière et al.). Lacks comparison to more recent Transcoder-IR (Szafraniec et al. 2022), which is stronger than Transcoder-ST.

**Justification For Why Not Lower Score:**

N/A

---

### Decision · Program_Chairs · 2024-01-16

Reject